# Effects of Climate Change Knowledge on Adolescents’ Attitudes and Willingness to Participate in Carbon Neutrality Education

**DOI:** 10.3390/ijerph191710655

**Published:** 2022-08-26

**Authors:** Jiaqi Zhang, Zepeng Tong, Zeyu Ji, Yuanchao Gong, Yan Sun

**Affiliations:** 1Key Laboratory of Behavioral Science, Institute of Psychology, Chinese Academy of Sciences, Beijing 100101, China; 2Department of Psychology, University of Chinese Academy of Sciences, Beijing 100049, China

**Keywords:** climate change knowledge, carbon neutrality education, environmental responsibility, mediating effect

## Abstract

The achievement of carbon neutrality has become increasingly important. Therefore, it the use of education to increase public understanding of carbon neutrality and facilitate low-carbon behaviors is urgent. Climate change knowledge is an effective measure to promote people’s interest and enthusiasm for specific educational projects. The present study analyzed the effects of climate change knowledge on adolescents’ attitudes and their willingness to participate in carbon neutrality education and validated the mediating effect of environmental responsibility. The findings showed that climate change knowledge improves adolescents’ attitudes toward carbon neutrality education and that environmental responsibility plays a mediating role in this. The findings provide insightful implications for carbon neutrality related policymaking and education promotion.

## 1. Introduction

The Chinese government has made addressing climate change a top priority, pledging to achieve peak carbon emissions by 2030 and carbon neutrality by 2060. However, carbon neutrality related legislation in China is still in its infancy. Public understanding of concepts, knowledge, skills, and carbon neutrality related policy remains relatively limited. Individuals’ participation in low-carbon behaviors is insufficient. According to 2020 data from the Ministry of ecological environment of China, for example, only 45.9% of survey respondents pay frequent attention to environmental information, and only 54.2% of respondents regularly sort waste in the household. Moreover, only 38.5% would consider purchasing green products. Therefore, encouraging the public to proactively participate in low-carbon behaviors is indispensable for the achievement of China’s carbon neutrality progress.

Environmental education effectively improves attitudes toward public environmental policies [1], as well as individuals’ willingness to participate in pro-environmental behaviors [2]. However, adults tend to have more stable behavioral patterns, consumer habits, and value orientations. In contrast, adolescents are more easily guided and subject to intervention. Adolescents begin to form early environmental affection, attitudes, and values in high school, and these have a far-reaching impact on the affective environmental attitudes and value orientation of the whole society [3]. Carbon neutrality education is a subset of environmental education. Past research shows that environmental education improves adolescents’ attitudes toward the environment [4,5], enhances their participation in climate issues [6], instills the notion of harmonious co-existence with nature, and helps to establish an appropriate ecological view of the world [7]. Carbon neutrality education deepens adolescents’ understanding of climate change and promotes a willingness to uphold pro-environmental behaviors. Therefore, it is a crucial measure that should be incorporated into carbon neutrality related legislation.

Regarding the current status of China’s carbon-neutral implementation, adolescents exhibit a relatively negative attitude and lack of willingness to participate in carbon neutrality courses. To some extent, this is related to the manner in which carbon neutrality education is currently promoted. Knowledge publicity and education are effective strategies for promoting educational activities. Publicity and education on environmental knowledge can increase adolescents’ willingness to participate in environmental knowledge and pique their interest in disseminating green concepts [8].

China’s carbon neutrality related legislation is still in an inchoate stage of development. Carbon neutrality education can be adopted to promote adolescents’ understanding of environmental policies and willingness to participate in pro-environmental behaviors. Therefore, this study examined the effects of climate change knowledge on adolescents’ participation in carbon neutrality education, providing insightful implications for improving public attitudes and willingness to achieve the carbon neutrality goals.

### 1.1. Climate Change Knowledge Popularization

Climate change knowledge popularization refers to themed group activities that serve as important media for improving public understanding of science. These activities are intended to popularize science’s technical knowledge and methods to a general audience [9]. Knowledge publicity and education for the youth involves science-themed activities that target adolescents, with education as the medium and the primary goal of improving adolescents’ scientific literacy [10].

Climate change knowledge popularization is divided into two main categories: classroom environmental activities and extracurricular climate change knowledge popularization. The former adopts a variety of active, student-centered teaching methods, including experiential, inquiry based, and constructivist approaches, which have all been proven to be effective in publicity and education on environmental knowledge [11,12]. The interventions featured in many papers comprise debates, group discussions, hands-on experiments, and field study teaching methods [4,5,13,14,15,16,17]. Whether these teaching methods are effective is also a significant research question. One study shows that, by incorporating music and graphical materials, climate change knowledge classes allow senior high school students to learn about climate change risks and recognize their own power to change the world [6]. An experiment after a climate change talk showed significant improvements in students’ climate science knowledge, climate issue participation, and climate-friendly behaviors that exceeded carrying a reusable water bottle to school. Those who were uninterested or skeptical about climate change science before the program were the most likely to pay increased attention to climate change and gain more knowledge about climate science [6]. Climate change knowledge classes also use videos to capture more interest. In Turkey, teachers who watched a documentary on climate change (An Inconvenient Truth) demonstrated significant learning gains, compared with a control group who did not watch it [18]. Similarly, an animated educational video shown to 10–11-year-old students in India helped raise interest and responses to global warming content [19]. Climate change knowledge classes also use role-play and simulations that mimic reality to help students understand various perspectives, predict events that could occur in the future, and increase their interest and enjoyment in learning [4,5]. Dresner [20] employed an energy conservation simulation in undergraduate classes and showed that simulating community participation to make a difference increased the of students’ willingness to do the same in the real world.

Extracurricular climate change knowledge popularization presents adolescents with the opportunity to initiate climate change projects at school or in the greater context of the community. In the United Kingdom, middle and high school students played a key role in reducing their schools’ energy use by gaining an understanding of energy resources, collecting data, monitoring energy use in the classroom, and planning and implementing appropriate action projects within their schools. After a few no-cost projects were implemented, their schools’ electricity use was reduced by 35% on average [21]. In several studies conducted in other countries, students participated in projects where they communicated climate change concepts to other people. In Crete, an energy efficiency education program engaged students in saving energy at home and in sharing information with their parents [22]. In the United States, university faculty developed a climate change course and assigned groups to produce public service announcements [23]. In an international exchange program, senior high school students collaborated on two service projects with individuals impacted by climate change, both of which influenced their level of concern toward climate change [24]. Upon returning to their own country, some students made active changes in their environmental behaviors, participated in various social action projects, and assumed the roles of school and community leaders who shared climate change information with other people. In Canada, students combined a research project with community outreach by conducting research on local climate change and presenting information to schools within their region in the form of a video [25]. The students planned and implemented several action projects, including a tree-planting project that used signage to explain the ecological value of trees. An interview with these students revealed that these projects contributed to changing their views toward climate change and had an empowering effect on them.

The above shows that developing a diverse and engaging mix of climate change knowledge popularization promotes adolescents’ participation in environmental education. Currently, most of China’s elementary and high schools have extracurricular interest groups. Competitions of various types are also frequently organized to lay the foundation for small-scale popular science bases for adolescents. By making full use of these bases, integrating them with elementary and high school students’ cognitive characteristics, and organizing educational activities in environmental science that popularizes resource and energy conservation, waste sorting and recycling, biodiversity, and low-carbon living, adolescents learn about the environment from a young age. This may draw increased attention to the environment and stimulate their interest in environmental protection while fostering their environmental awareness and pro-environment behaviors [8].

Engaging in climate change knowledge popularization inspires adolescents to contemplate the inextricable relationship between carbon neutrality and each individual, achieving long-term education by fostering cascading changes from the student to the family, community, and beyond. Previous findings also indicate that students can play a substantial role in new energy development. Promotional campaigns in the field of popular science, including Earth Day and Arbor Day, have both elevated existing student knowledge and broadened their scope. These activities prompt students to closely integrate social reality with theories and material learned in class, thereby increasing the depth of their scientific knowledge [26].

In light of this, engaging adolescents in climate change knowledge popularization is an important means to increase low-carbon behaviors and achieve carbon neutrality in the future. This study’s fundamental purpose for selecting adolescents as research targets was to enhance the stable and sustained development of carbon neutrality. It is reasonable to assume that engaging in climate change knowledge popularization motivates adolescents to proactively participate in carbon neutrality education. Thus, this study hypothesized the following:

**H1:** 
*Climate change knowledge increases adolescents’ supportive attitudes toward carbon neutrality education, willingness to participate in related courses, and the preferred hours of instruction.*


### 1.2. Environmental Responsibility

Environmental responsibility refers to individuals’ belief that they should take action to mitigate environmental issues because they are fully aware of the environmental gains and losses associated with their own behaviors. Comprising environmental issue perceptions, environmental awareness, and willingness to participate in environmental behaviors, it represents individuals’ self-accountability to forgo their own economic gains from consumption and recognize and internalize social norms. It also reflects mental qualities, such as the courage, resilience, self-restraint, and selflessness involved in rectifying environmental issues. Thus, this sense of responsibility will be a powerful driving force that motivates individuals to assume environmental responsibility and adopt pro-environmental behaviors [27].

Environmental responsibility is measured with a scale that comprises four items. The measurement dimensions encompass concern for environmental issues, environmental awareness, and a willingness to contribute to the environment. Environmental awareness is measured by items such as “It is difficult for people like me to do anything to protect the environment” [28,29]. Environmental responsibility is also manipulated in various ways. In Wu and Yang’s [30] experiment, scholars studied data on municipal solid waste pollution and individual waste generation estimates published by the Chinese Ministry of Ecology and Environment. These scholars were also presented with related graphs and texts to stimulate their environmental responsibility. Afterward, they were asked to recall their past experiences and write down three incidents where environmental damage was caused by their behavior [27]. Thus, this study hypothesized the following:

**H2:** 
*Climate change knowledge promotes environmental responsibility.*


The Model of Responsible Environmental Behavior postulates that individual environmental responsibility is an important cognitive driver of pro-environmental behaviors. Individuals with a strong sense of environmental responsibility are more likely to display pro-environmental behaviors. Although the positive influence of environmental responsibility on individuals’ environmental behaviors was empirically validated by Stern et al. [29], views are divided regarding whether this influence is exerted via a direct or indirect pathway. In later studies, most scholars report that environmental responsibility’s effect on environmental behaviors is indirect and mediated or moderated by other factors. Discussions on the mediating factors have focused largely on public perception and involved concepts such as personal values. Public environmental responsibility significantly and positively influenced environmental behaviors through subjective norms, consumer values, and moral identity [31]. Individuals who hold themselves accountable are more likely to behave in a pro-environmental way. Thus, this study hypothesized the following:

**H3:** 
*Environmental responsibility motivates adolescents to proactively participate in carbon neutrality education.*


Environmental responsibility entails perceptions of the environment’s current state, environmental awareness, and environmental behavioral awareness. As a psychological driver, it guides and motivates consumers to actively assume responsibility for environmental issues, internalize social norms as subjective norms, and make concrete efforts to help improve the environment. Consumers are also encouraged in highlighting mental qualities such as courage, devotion, resilience, self-restraint, and selflessness by adopting green behaviors, reducing their environmental footprint, and addressing environmental issues.

Individual environmental responsibility is an essential cognitive driver of pro-environment behaviors; therefore, individuals with a greater sense of environmental responsibility are more likely to adopt them. As these individuals are also more enthusiastic about engaging in environmental education, environmental responsibility predicts a willingness to participate in carbon neutrality education. Hence, this study hypothesized the following:

**H4:** 
*Environmental responsibility mediates the impact of climate change knowledge on carbon neutrality attitude and willingness.*


### 1.3. Current Study

The study hypotheses offer potential interventions for carbon neutrality education. In addition, the study tested environmental responsibility’s value as a mediating variable to provide valuable guidance for carbon neutrality education for adolescents. Figure 1 shows the study’s theoretical model.

## 2. Methods

### 2.1. Participants

This study comprised junior high school students from Beijing, China as participants and conducted a survey in November 2021 and 3005 questionnaires were distributed. We removed the ineligible questionnaires with missing values and finally obtained 2964 valid samples. These samples are a good representation of the overall situation of middle school students in Beijing. Participants comprised 1469 males (49.6%) and 1495 females (50.4%), with an average age of 12.70 (±0.40) years. Household registration showed that 93.8% resided in urban areas. Parents’ professions included only 20.6% who worked in energy related industry. The demographic characteristics of the participants are shown in Table 1.

### 2.2. Materials

The questionnaire consists of cognitive indicators, behavioral indicators, and environmental psychological indicators. All the contents of the questionnaire can be seen in Table A2 in the Appendix A. The investigation was carried out as follows: the survey was conducted by the corresponding teacher of each class during psychology sessions. The teachers announced the instructions beforehand and collected the questionnaires at the end of the session. Each interviewer attended a brief training session for ensuring the accuracy of survey results.

#### 2.2.1. Climate Change Knowledge

The carbon neutrality knowledge and behaviors questionnaire comprised four items that reflected the current organization of climate change knowledge, such as “Has your school carried out poster design activities on energy conservation, emission reduction, low carbon and environmental protection?” Participants were asked to respond on a scale from 1 to 3 (1 = “yes”; 2 = “no”; 3 = “no idea”). The “no idea” option was included to avoid untruthful responses from participants who were forced to answer. It was processed in the same manner as the “no” response in the final statistical analysis.

#### 2.2.2. Environmental Responsibility

The carbon neutrality knowledge and behaviors questionnaire comprised five items that reflected participants’ environmental responsibility, such as “Everyone is responsible to take action in protecting the environment”. Using an 11-point scale, the participants rated themselves from 0 = “extremely disagree” to 10 = “extremely agree”, where a higher score denoted a stronger sense of environmental responsibility. The Cronbach’s α coefficient for the environmental responsibility subscale was 0.804 in this study.

#### 2.2.3. Willingness to Participate in Carbon Neutrality Education

The carbon neutrality knowledge and behaviors questionnaire comprised three indicators measured across five items that reflected willingness to participate in carbon neutrality education. The willingness to participate in related courses was measured by the item “I am very interested in taking courses on low carbon environmental protection.” Participants rated themselves on an 11-point scale (0 = “extremely disagree”; 10 = “extremely agree”), where a higher score indicated more interest and greater willingness to participate in carbon neutrality courses.

The preferred hours of instruction were measured by the item “If allowed, I expect to dedicate [0–5] class hours to education on low-carbon environmental protection every week”. Supportive attitudes toward education were measured by three items such as “I think it is absolutely necessary that the school offers low-carbon environmental education”. Higher scores reflected a more positive attitude toward carbon neutrality education. The Cronbach’s α coefficient in this study was 0.847.

### 2.3. Data Analysis

IBM SPSS v 26.0 was used to conduct descriptive statistics, independent samples *t*-test, and regression analysis. The SPSS PROCESS macro version 3.3 (Model 4) [32,33] tested the mediating effect of environmental responsibility. The statistical significance of the indirect effect of the mediator was estimated using 5000 bootstrap samples to construct a 95% bias-corrected and accelerated (BCa) confidence interval (CI). If zero was not included in the 95% CI, the indirect effect was significantly different from zero at *p* = 0.05.

## 3. Results

### 3.1. The Effects of Climate Change Knowledge on Willingness to Participate in Carbon Neutrality Education

The tables in Appendix A show the results of the independent samples t-test on the effects of four types of climate change knowledge on supportive attitudes toward education, willingness to participate in related courses, and preferred hours of instruction.

The result shows that (see Appendix A, Table A2) participants who did not participate in any of the four activities on popularizing knowledge about climate change were significantly less supportive of carbon neutrality education and were less willing to participate in carbon neutrality courses than those who had participated in producing handwritten newspapers, donating used goods, popular science fairs, or popular science talks. In addition, the former also preferred significantly fewer hours of carbon neutrality education than those who had participated in the activities: producing handwritten newspapers (t = −3.92, *p* < 0.001), donating used goods (t = −3.17, *p* = 0.002), and popular science fairs (t = −3.27, *p* < 0.001).

It can be inferred from Table A3 in Appendix A that producing handwritten newspapers, popular science fairs, and popular science talks all had a significant effect on participants’ supportive attitudes toward carbon neutrality education. The largest effect size was observed for popular science fairs (β = 0.14), followed by popular science talks (β = 0.12), and producing handwritten newspapers (β = 0.10).

The result in Table A4 in Appendix A shows that producing handwritten newspapers, popular science fairs, and popular science talks had a significant effect on adolescents’ willingness to participate in carbon neutrality courses. The largest effect size was observed for popular science fairs (β = 0.114), followed by producing handwritten newspapers (β = 0.084), and popular science talks (β = 0.078). 

Producing handwritten newspapers and donating used goods both had a significant effect on the preferred hours of carbon neutrality education among adolescents (see Table A5 in Appendix A). The largest effect size was observed for producing handwritten newspapers (β = 0.07), followed by donating used goods (β = 0.04).

### 3.2. Effects of Climate Change Knowledge on Environmental Responsibility

The independent samples’ *t*-tests show that participants who did not participate in any of the four climate change knowledge popularization activities reported a significantly lower sense of environmental responsibility than those who had. Our regression analysis revealed that producing handwritten newspapers and popular science fairs both had a significant effect on adolescents’ environmental responsibility. A higher level of participation was associated with a greater sense of environmental responsibility. Between the two activities, donating used goods had a greater effect on environmental responsibility.

### 3.3. Mediating Effect of Environmental Responsibility

The SPSS PROCESS macro version 3.3 (model 4) [32,33] was applied to test the mediating effect of environmental responsibility, as shown in Figure 2, Figure 3 and Figure 4.

The result shows that the regression coefficient of climate change knowledge to environmental responsibility is 0.19 and the regression coefficient of environmental responsibility to supportive attitudes toward carbon neutrality education is 0.65, and the regression coefficient of climate change knowledge to supportive attitudes toward carbon neutrality education is 0.50, as shown in Figure 2. The result shows that climate change knowledge had a total effect of β = 0.50 and a direct effect of β = 0.38, and the mediating effect of environmental responsibility was β = 0.12, as shown in Appendix A, Table A6. Significant total, direct, and mediating effects were observed for all four of the climate change knowledge, both individually and collectively (*p* < 0.01). In addition, the results remained valid after gender, age, household registration, and parents’ profession were incorporated into the model as control variables.

Results of Figure 3 show that the regression coefficient of climate change knowledge to environmental responsibility is 0.19 and the regression coefficient of environmental responsibility to willingness to participate in carbon neutrality courses is 0.15, and the regression coefficient of climate change knowledge to willingness to participate in carbon neutrality courses is 0.06.The result shows that (see Appendix A, Table A7) climate change knowledge had a total effect of β = 0.46 and a direct effect of β = 0.34, and the mediating effect of environmental responsibility was β = 0.13; producing handwritten newspapers had a total effect of β = 1.07 and a direct effect of β = 0.79, and the mediating effect of environmental responsibility was β = 0.29; donating used goods had a total effect of β = 0.53 and a direct effect of β = 0.27, and the mediating effect of environmental responsibility was β = 0.26; popular science fairs had a total effect of β = 1.31 and a direct effect of β = 1.02, and the mediating effect of environmental responsibility was β = 0.30; popular science talks had a total effect of β = 1.21 and a direct effect of β = 0.89, and the mediating effect of environmental responsibility was β = 0.33. Significant total, direct, and mediating effects were observed for all four of the climate change knowledge, both individually and collectively (*p* < 0.01). In addition, the results remained valid after gender, age, household registration, and parents’ profession were incorporated into the model as control variables.

From the results in Figure 4, we can see that the regression coefficient of climate change knowledge to environmental responsibility is 0.19 and the regression coefficient of environmental responsibility to preferred hours of carbon neutrality education is 0.15, and the regression coefficient of climate change knowledge to preferred hours of carbon neutrality education is 0.06. It can be inferred from Table A8 in Appendix A that climate change knowledge had a total effect of β = 0.08 and a direct effect of β = 0.06, and that the mediating effect of environmental responsibility was β = 0.03; producing handwritten newspapers had a total effect of β = 0.26 and a direct effect of β = 0.20, and the mediating effect of environmental responsibility was β = 0.06; donating used goods had a total effect of β = 0.21 and a direct effect of β = 0.15, and the mediating effect of environmental responsibility was β = 0.06; popular science fairs had a total effect of β = 0.19 and a direct effect of β = 0.13, and the mediating effect of environmental responsibility was β = 0.06; popular science talks had a total effect of β = 0.09 and a direct effect of β = 0.01, and the mediating effect of environmental responsibility was β = 0.07. Significant total, direct, and mediating effects were observed for all four of the climate change knowledge activities, both individually and collectively (*p* < 0.01). The results remained valid after gender, age, household registration, and parents’ profession were incorporated into the model as control variables.

## 4. Discussion

### 4.1. Facilitating Effects of Climate Change Knowledge

This study revealed that climate change knowledge has a positive effect on adolescents’ willingness to participate in carbon neutrality education. This effect was observed in adolescents’ supportive attitudes toward education, willingness to participate in related courses, and preferred hours of instruction. The finding offers insightful implications regarding public attitudes and willingness to achieve the national peak carbon emissions and carbon neutrality goals.

The findings also showed that producing handwritten newspapers, donating used goods, popular science fairs, and popular science talks were all effective in promoting adolescents’ supportive attitudes toward carbon neutrality education and willingness to participate in related courses. With the exception of popular science talks—which had no significant effect on the preferred hours of instruction—the three other activities increased the hours of carbon neutrality education in which adolescents were willing to participate. Previous research also highlights differences in the effectiveness of various teaching methods [34]. Similar to adolescents’ schooling, popular science talks are relatively consistent with the lectures delivered by teachers. Therefore, such talks had no significant effect on the preferred hours of carbon neutrality education and were less experiential than the other three activities.

The regression analysis results showed that all four climate change knowledge popularization activities enhanced supportive attitudes toward carbon neutrality education. However, willingness to participate in related courses was only promoted by producing handwritten newspapers, popular science fairs, and popular science talks. While producing handwritten newspapers and donating used goods increased the preferred hours of carbon neutrality education among adolescents, popular science talks and fairs had no significant effect. It is also worth noting that producing handwritten newspapers exerted the most significant effect on supportive attitudes toward education, willingness to participate in related courses, and preferred hours of instruction. Previous research has shown that providing climate change content through inquiry based activities more effectively promotes student learning. These activities allow students to develop their own knowledge and draw conclusions based on this knowledge [35]. Svihla and Linn [36] discovered that inquiry based activities helped high school students to better understand and integrate the knowledge obtained from interactive visualizations and make decisions pertaining to energy use. McNeal et al. [37] reported that inquiry based activities not only increased students’ conceptual understanding, but also improved their understanding of how complex systems interact. They also found that improved learning outcomes were most significant in classes that completed the entire set of labs. Producing handwritten newspapers was more experiential because it required students to process and reconstruct the knowledge they learned, thereby deepening their understanding and recognition of such knowledge through the inquisitive process. Students were also required to apply their creativity to the design process. Hence, this had the most significant effect on supportive attitudes toward carbon neutrality education, willingness to participate in related courses, and preferred hours of instruction.

### 4.2. Mediating Effect of Environmental Responsibility

Climate change knowledge enhances environmental responsibility, which plays a role in promoting courses and activities related to carbon neutrality education. The four types of climate change knowledge activities motivated adolescents to proactively participate in carbon neutrality education, both individually and collectively. Environmental responsibility plays a mediating role and casts light on the potential psychological mechanism underlying the effects.

All four activities led to participants’ enhanced sense of social responsibility. Producing handwritten newspapers and popular science fairs both promoted environmental responsibility, and donating used goods had a greater effect on environmental responsibility. According to previous research, public environmental responsibility significantly and positively influences individuals’ environmental behaviors through subjective norms and moral identity. Individuals who hold themselves accountable are more likely to behave in a pro-environmental way. As a psychological driver, environmental responsibility guides and motivates consumers to assume active responsibility for environmental issues, internalize social norms as subjective norms, and make concrete efforts to help improve the environment [29]. Thus, climate change knowledge allows individuals to both obtain environmental knowledge and gain a deeper understanding of the environment. They also reinforce individuals’ identity as a “low-carbon” and “environmentally-friendly” person, causing them to hold themselves more accountable for the environment. Among the four activities, donating used goods had the greatest effect because it was the most charitable and was therefore effective in generating a sense of responsibility.

An abundance of existing evidence attests to the pivotal role of climate change knowledge in improving attitudes toward and willingness to participate in education. Proponents of environmental protection tend to be younger [38,39], because adolescents are predisposed to accept new ideas [40,41]. Carbon neutrality is an emerging concept. While previous research focuses on climate change, no previous studies have investigated carbon neutrality education for adolescents. As engaging in climate change knowledge facilitates low-carbon behaviors, these activities are important for future environmental education. Engaging youth groups in climate change knowledge will address existing environmental issues and play a major role in facilitating future environmental protection.

### 4.3. Implications for Managerial Practices

Our findings offer the following implications for carbon neutrality education for adolescents. First, organizing projects to produce handwritten newspapers on carbon neutrality at school can motivate adolescents to proactively participate in carbon neutrality education. Schools can strengthen the collective environmental responsibility among adolescents by sharing carbon neutrality knowledge and hosting related talks. Second, science and technology museums and parks should initiate environmental education projects on carbon neutrality topics and regularly produce and present programs for youth groups to promote low-carbon behaviors. The China Science and Technology Museum, for example, organized the “30.60” Peak Carbon Emissions and Carbon Neutrality Themed Exhibition with interactive games, such as carbon footprint tracking and waste sorting simulations, which received sound results. Additionally, Beijing Wenyuhe Park built the Future Wisdom Valley, a park space abounding with carbon-neutral interactive facilities that support interactive games, such as bicycles that can track the amount of carbon emissions saved by the cyclist. Finally, environmental education targeting teachers and parents is also indispensable. Courses that enhance their skills and approaches to environmental education should be offered regularly. This will help them to better educate children on carbon neutrality, foster the proper environmental values, and improve their environmental responsibility, thereby enhancing their willingness to participate in carbon neutrality education.

### 4.4. Limitations and Future Directions

The selection of research participants had limitations. The study enrolled participants primarily from Beijing junior high schools, with only a few from other regions. The lack of diversity may have resulted in the findings being influenced by other factors, such as geographical location, school, and grade, which would undermine the representativeness and commonality of the results. Furthermore, this study is based on an online survey in which all variables were self-reported. This method carries potential risks from social expectations [42]. As a portion of the questionnaires were first distributed to participants’ class teachers, the actual results may be affected by factors related to social praise, self-image management, and personal values.

Future studies should focus on participants at different school levels. Stratified sampling can improve the overall reliability and validity, resulting in more profound and comprehensive research. Second, future studies may consider controlling for pro-environmental behavioral tendencies related to the questionnaire method. Alternative experimental approaches can be adopted to partially obscure the purpose of the experiment to better explore participants’ genuine willingness to participate in carbon neutrality education. Finally, because temporal continuity cannot be achieved through the questionnaire method, future studies can perform continuous observations at different stages.

It is recommended that future researchers expand the sample to a nationwide group of students and break the limit of junior high school students. Beijing is one of the most developed regions in China in terms of educational resources and places great emphasis on education related to carbon neutrality. Based on data from a large sample of students in Beijing, this study finds that climate change knowledge improves adolescents’ attitudes toward carbon-neutral education, and that environmental responsibility plays a mediating role in this. Currently, the penetration of carbon neutral education in schools nationwide is still low. Therefore, the findings of this study are leading and instructive for carbon neutral education nationwide. This suggests that future researchers can use this study as a basis to further explore the generalizability of the experimental findings to other regions and other groups.

## 5. Conclusions

In conclusion, this study found that a higher level of climate change knowledge can motivate adolescents to proactively participate in carbon neutrality education, and this positive effect is mediated by environmental responsibility.

## Figures and Tables

**Figure 1 ijerph-19-10655-f001:**
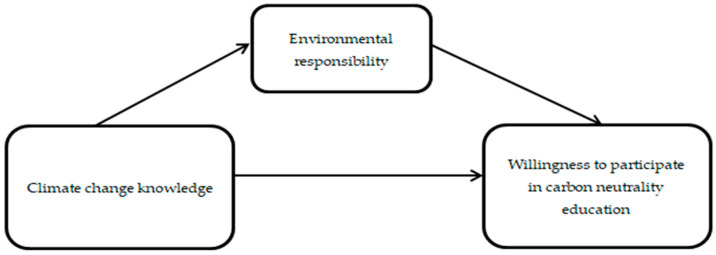
Theoretical Model.

**Figure 2 ijerph-19-10655-f002:**
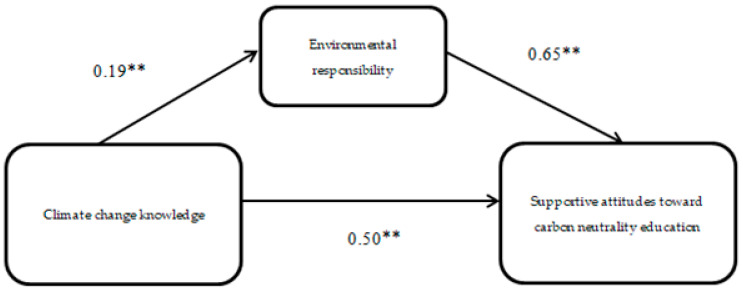
Model of supportive attitudes toward carbon neutrality education. Note: ** *p* < 0.01.

**Figure 3 ijerph-19-10655-f003:**
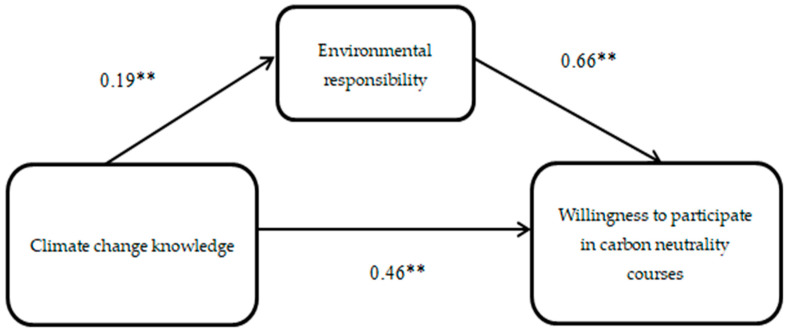
Model of willingness to participate in carbon neutrality courses. Note: ** *p* < 0.01.

**Figure 4 ijerph-19-10655-f004:**
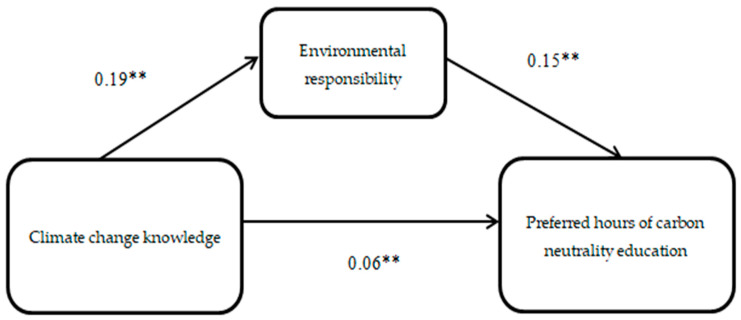
Model of preferred hours of carbon neutrality education. Note: ** *p* < 0.01.

**Table 1 ijerph-19-10655-t001:** Participants demographic characteristics (*N* = 2964).

Demographic Variable	*N*	%
Gender		
Male	1469	49.6
Female	1495	50.4
Age		
8–11	235	7.9
12–15	2605	87.9
16–22	124	4.2
Household registration		
Urban	2781	93.8
Rural	183	6.2
Nature of parents’ profession		
Related to energy industry	611	20.6
Unrelated to energy industry	1967	66.4
No idea	386	13.0

## Data Availability

All data were uploaded on the Figshare. Other researchers can download the dataset at https://doi.org/10.6084/m9.figshare.20464338.v1 (accessed on 6 August 2022).

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
