# Peer review of "Effects of Climate Change Knowledge on Adolescents’ Attitudes and Willingness to Participate in Carbon Neutrality Education"

_ijerph, 2022, doi:10.3390/ijerph191710655_

Round 1
Reviewer 1 Report
Carbon Neutrality is a hot topic in current research. In order to realize such a goal, it is important to know adolescents' knowledge about it. Thus this article is selecting an interesting topic. However, the following issues much be improved:
1. Please make clear how the questionnaires were designed; why the number of valid questionnaires were reliable; how the questionnaires were distributed in the study area; why each kind of the questionnaires represented the whole characteristics, for example, whether the proportions of questionnaires for male and female were similar to the proportions of male and female in the whole study area.
2. Lines 205-209, why do you name them as H1-H3?
3. Please put the contents of questionnaires in Table.
4. part 2.2.3, why do you divided the scale into 11 points? Do you have solid foundation?Why not use 1-5 Likert scale?
5. In Tables 4 and 5, other factors of respondents' socio-economic characteristics should be analyzed and discussed.
6. Tables should be mentioned in results analyses. For example: It can be inferred from Table * that....
Reviewer 2 Report
This paper while interesting is far too long. There is significant scope to reduce the length of the paper, without loosing the message.
I have some concerns with table 1, while you have 3005 participants listed gender is only known for 2964 participants, perhaps the analysis should be restricted to those whose gender is known, particularly as you undertake a gender analysis of the data. Also the table is not consistent with the text in the article.
Tis study was undertaken in a particular location, with unique circumstances, how transferable the knowledge is to other regions is not well discussed.
I think Many of the tables could be provided in an appendix without the loss of the message.
